# Dog-Assisted Therapy and Dental Anxiety: A Pilot Study

**DOI:** 10.3390/ani9080512

**Published:** 2019-07-31

**Authors:** Norma Cruz-Fierro, Minerva Vanegas-Farfano, Mónica Teresa González-Ramírez

**Affiliations:** 1School of Dentistry, Autonomous University of Nuevo León (Facultad de Odontología, Universidad Autónoma de Nuevo León), Nuevo León C.P. 64455, Mexico; 2School of Sports Management, Autonomous University of Nuevo León (Facultad de Organización Deportiva, Universidad Autónoma de Nuevo León), Nuevo León C.P. 64455, Mexico; 3School of Psychology, Autonomous University of Nuevo León (Facultad de Psicología, Universidad Autónoma de Nuevo León), Nuevo León C.P. 64455, Mexico

**Keywords:** anxiety, dental anxiety, dog-assisted therapy

## Abstract

**Simple Summary:**

The participation of animals, specifically dogs, in therapeutic activities has been demonstrated to improve individuals’ physical and mental health. However, few investigations have been carried out in the area of dentistry. This study was conducted to evaluate the effect of dog-assisted therapy for individuals with a history of anxiety related to dental visits. During preventative dental procedures (cleaning), a therapy dog accompanied the participant. After this intervention, people reported lower perceived discomfort at dental visits. The participation of therapy dogs in this area of health could help improve people’s experiences in dental offices.

**Abstract:**

Animal-assisted therapy aimed at improving individuals’ mental or physical health has been widely reported. However, the data on how a therapy dog could help control anxiety during dental procedures is scarce. The objective of this work was to evaluate the effect of dog-assisted therapy on people with a history of dental anxiety while receiving preventive dental treatment. Twelve adults participated (women: n = 11 (91.7%) and men: n = 1 (8.3%), mean age = 31.25 years, D.E. = 5.78). The Corah Dental Anxiety Scale was applied, the patient’s mood was assessed with a Likert scale before and after receiving the dental treatment, and their blood pressure was recorded for each of the three stages of treatment. A therapy dog accompanied the participants during the dental procedure. The main results indicated that a decrease in discomfort was perceived during the intervention, and there was also an improvement in the patient’s evaluation of the experience. The results are based on the decrease in patients’ blood pressure when taken in the middle of the dental treatment.

## 1. Introduction

Fearing or refusing dental treatment is a negative emotional reaction manifested by cognitive, physiological, and motor responses [1], which, when associated with dental visits or dental treatment, is referred to as dental anxiety [2].

Avoidance behavior is one of the main causes of canceled appointments or lack of interest in dental treatment, which in turn generate tooth decay, decreased self-esteem, and poor quality of life [3,4]. However, the responses observed in each person vary depending on individual coping resources [5,6]. This variation in response, together with its multifactor origin [7], contributes to the lack of clarity in its etiology, making it difficult to choose the most appropriate treatment for its control [1].

Dental anxiety can be acquired through direct conditioning, generating mostly physiological and behavioral responses, or through indirect conditioning through experiences narrated by other people, tending to prompt subjective responses [1,8,9,10,11].

To control these negative reactions during dental treatment, different techniques have been proposed, which are mainly focused on non-pharmacological management (tell–show–do technique, voice control, non-verbal communication, positive reinforcement, distraction), pharmacological management with nitrous oxide, sedation, and general anesthesia [12], or modifying the environment through both sounds and lights specifically designed for helping pediatric patients [13].

In addition to these traditional techniques, the use of animal-assisted therapy (AAT) has been proposed as an alternative that produces calming effects for people, as previously learned from relationships between humans and animals [14]. This finding is supported by results that have been reported in animal-assisted activities (AAA) in different areas, such as visits to hospitals and nursing homes; in interventions to support at-risk youth or delinquents; and in psychotherapy, social work, and physiotherapy, as well as in reading and education programs [15,16].

There is broad historical support for these activities in the medical field, beginning with the work of Nightingale in 1800 [17]. In 1962, Levinson formally integrated animal-assisted activities with clinical psychology [18]. Since then, different studies have confirmed their effects, and to date, this form of therapy has been supported by different organizations worldwide [19].

Currently, in different parts of the world, beneficial effects have been observed in patients visiting hospitals accompanied by therapy dogs, with reports of their influence on improving blood pressure and neurohormone levels and facilitating anxiety management [20,21]. Although these hospital activities follow strict clinical guidelines, such as those proposed by the Society for Healthcare Epidemiology of America [22], their recent inclusion and the cultural differences regarding the role of animals limit the dissemination of such practices in various areas.

In dental practice, various publications have reported that dog-assisted activities contribute to reducing anxiety, among which the report by Solana [23] for the American Dental Association (ADA) stands out, as does the study by Cajares, Rutledge, and Haney [24], who measured the levels of dental anxiety in a group of people with intellectual and developmental disabilities. Studies have confirmed that these animal-assisted activities cause a relaxing effect in people that allows them to divert their attention to the dog and therefore reduce their anxiety [25].

It is important to expand the data to strengthen alternative techniques that contribute to prevention, thereby improving individuals’ oral health and quality of life and expanding knowledge of the benefits obtained through AAT as a support method for the management of dental anxiety.

The purpose of this study was to evaluate changes in dental anxiety when introducing a therapy dog during a prophylactic procedure, hypothesizing a decreased state of anxiety in the participants given the presence of the canine assistant during the intervention.

## 2. Materials and Methods

A non-probabilistic convenience sampling was conducted whose inclusion criteria required being of legal age, having expressed apprehension (stress, anxiety, fear) when going to the dentist, and liking dogs.

The following were considered exclusion criteria: fear of interacting with dogs, regardless of their size; disgust at being licked by dogs; allergic to dogs.

Instruments: The Corah Dental Anxiety Scale (CDAS) was used [26], which is a Likert-type instrument with four questions scored from 1–5 points for a total possible value of 20. A score of 13 or 14 suggests to the dentist that the patient has dental anxiety, and a score of 15 or more generally indicates a patient with high dental anxiety [27]. For this study, a minimum score of 13 was considered indicative of anxiety sufficient for inclusion in the study; the reliability of the instrument had a value of α = 0.73.

To identify possible differences between the participants’ prior experiences and what was experienced during this session, a questionnaire was designed with two questions aimed at prior experiences with dentistry and three at their participation in the project assisted by a dog. The following questions were asked prior to the intervention: “How did you feel about going to the dentist?” and the open-ended “Going to the dentist is…” After the office visit, the following questions were asked: “How do you feel?”, “Going to the dentist is…”, and “What did you think of going in with (name of the dog)?” The questionnaire was designed as a Likert scale with six possible answers, the numerical values of which were weighted from 1 (the worst) to 6 (the best). The reliability of the instrument had a value of α = 0.67. Item analysis revealed that the last question, which was oriented to how the patient perceived having gone in with the animal, showed low item-total correlation; after this item was eliminated, instrument reliability increased to α = 0.70.

We used a Beurer BC-58 wrist blood pressure monitor, Beurer GmbH, Neu-Ulm, Germany, which performs an automatic measurement of arterial pressure (systolic and diastolic pressure) and radial pulse, following the manufacturer’s instructions for use.

Therapy dogs: For this study, the participation of four dogs, three females and one male, was included; the dogs’ breeds were English Shepherd, Schnauzer, Border Collie, and Labrador Retriever. All the dogs were adults, sterilized, vaccinated, and dewormed, with an average age of 40 months (SD = 19.25). Of these, two dogs already had participated in various workshops and activities as therapy dogs; for the other two, this experience was their first. All the dogs were evaluated by a canine trainer certified by the Certification Council for Professional Dog Trainers (CCPDT) for selection considering the circumstances and environment under which they would work: reduced space, new visual and auditory stimuli, and close contact with unknown people, among others. Likewise, a test session was conducted in place with each dog separately before being included in the study.

Canine teams worked as volunteers; handlers were not compensated for participating, and were also previously trained in the privacy standards and given all the necessary details about the dental treatment.

The research project adheres to ethical standards and was approved by the Bioethical Review Commission of the School of Dentistry of the Autonomous University of Nuevo León on 27 May 2016, sheet 0100 and code SPSI-010613.

Procedure: The invitation to participate in this research project was publicized on a page within Facebook conducted by researchers of a research group of the School of Psychology, which was oriented to the dissemination of scientific information on the human–animal bond. The invitation included the objective of the research, and its procedure was reported: participating in a preventive conventional dental prophylactic treatment without cost. Contact was made via email. The interested parties received more detailed information about the project, which included the previously described inclusion criteria, the dates and time available, which depended on the availability of the volunteer handlers, and the need to sign an informed consent once the project was made known in greater detail. The form authorized the indicated procedure and permitted videotaping the session, and the participant agreed to complete the previously described questionnaires.

Once the day of the appointment was agreed upon, a photograph of the therapy dog that would assist the individual was sent to the person. Dogs were randomly assigned to patients, with the sole criterion that both coincided in date and time. On the appointed date, the therapy dog and its handler were introduced, and instructions were given regarding the treatment of the dog (where and how to hold it, so that the dog feels comfortable) and telling the participant that he or she should feel free to touch or caress the dog when he or she felt anxious and/or stressed. Blood pressure and pulse were taken at three periods: before, midway through treatment, and at the end of treatment.

The questions aimed at knowing their previous and current experiences regarding the intervention were delivered before and at the end of the intervention.

The guidelines for the basic measures for the prevention of biological risks, disinfection of surfaces and instruments, and the use of disposable barriers and supplies, according to the Official Mexican Standard, NOM-013-SSA2-2006 [28], were followed.

The sessions were carried out following an infection control protocol, using physical barriers to avoid direct contact of the dog with the furniture. The equipment was used exclusively for the intervention. For infection control, before and after the session, the areas were thoroughly cleaned to remove organic material, followed by basic disinfection, using a 1:32 dilution of household bleach (1/2 cup of sodium hypochlorite per gallon of water). Periodically, the areas were fumigated [29,30].

For the disinfection of surfaces and prevention of zoonoses, the guidelines recommended in the compendium of measures to prevent diseases associated with animals in public places were followed [31], as were guidelines for hospitals [32,33,34].

The dogs’ health was monitored by the veterinarian, who certified that the dog was healthy, had a complete vaccination and deworming program in place, and showed no signs of physical, dermatological, or oral diseases. The dog’s temperament was evaluated by the trainer, as previously mentioned, and handlers were in frequent contact with her. Patient contact with any bodily fluid was avoided. In addition, prior to the appointment, the dog was brushed to remove loose hair. At all times, the dog’s behavior and welfare was supervised by its handler [35].

During the intervention, the dog was placed on a clean towel over the patient’s legs; both at the beginning and at the end of the intervention, the participant was invited to wash and sanitize their hands [22,36]. The dentist was assisted by a psychologist who monitored the patient’s behavior and well-being, took readings from the blood pressure monitor, and provided any necessary supplies, thus preventing the dentist from abandoning his or her activities and thereby compromising the hygiene of the procedure.

Statistical analysis: The reliability of all the questionnaires was analyzed before performing any other analysis. To determine the participants’ history of dental anxiety, descriptive measures of the mean and standard deviation of the CDAS scale were taken. To evaluate changes in dental anxiety, the responses to the questionnaire designed for this study were analyzed using the Wilcoxon rank test. The values of heart rate, high blood pressure, and low blood pressure were contrasted relative to three periods—before, midway, and end of the intervention—with the Friedman test.

## 3. Results

Twelve people with a history of dental anxiety participated in this study, all of whom were adults living in the metropolitan area of Nuevo León, Mexico. Of these, 11 were women (91.7%), and one was a man (8.3%). The mean age of the group was 31.25 years (SD = 5.78). Three of them, 40%, were married; the rest were single. Regarding the level of dental anxiety, the group had an average score on the CDAS scale of 14.41 points (SD = 2.74).

Perceptions and feelings prior to the intervention, which related to the degree of dental anxiety that each of these items aroused, are presented in Table 1. As observed, all the items had a mean score greater than or equal to the mean level of the scale, which was scored as tension (value 3) and close to anxiety (value 4). The average value of the summed results of the scale was 14.41 (SD = 2.74). The frequency and percentage per question of each of the questionnaire items are shown in Appendix A.

As for the questions oriented to the participant’s perceptions and feelings before and after the assisted dental intervention with dogs, a significant difference was obtained between how the participant felt in previous visits (How did you feel about going to the dentist?) and the current session (How do you feel?): Z = 2.96, *p* = 0.003. The average value of how they felt in previous visits was 3.2 (range = 4), and the average value of the current visit was 4.9 (range = 3). That is, an improvement was reported after the dog-assisted intervention. In relation to how both experiences were perceived, significant differences were also found: Z = 2.89, *p* = 0.004. In this second category, the mean value of the perception prior to the dentist visit (Previsit question: Going to the dentist is…) was 2.2 (range = 4); after dog-assisted intervention (Postvisit question: Going to the dentist is…), it was 4.5 (range = 3) (see Table 2).

Finally, Table 3 shows the descriptive data and comparison tests of the three physiological measures taken before, during, and after the intervention. There is a significant difference in high and low blood pressure, with a decrease during the intervention (midpoint), i.e., when accompanied by the dog. While pulse also decreases, this change does not reach statistical significance.

The analysis of the pulse and blood pressure measurements at the beginning and midpoint of the intervention showed that in all three measurements, these had decreased during the assisted intervention with the dog (Table 4). It is deduced that the presence of the dog helped the person feel more relaxed in the middle of the procedure, while the increase observed at the end of the procedure could be because the person was reactivating to get up from the chair and leave the doctor’s office, since the treatment had ended.

## 4. Discussion

This study is based on the exploration of the potential of assisted interventions with animals in the field of dental health through the focus on psychological phenomena. As its sole objective, the evaluation of the effect of canine accompaniment in dental practice for the reduction of dental anxiety was proposed, and it was hypothesized that the presence of the dog would decrease participants’ anxiety.

To achieve this effect, participants with a history of dental anxiety verifiable through a reliable psychometric evaluation were included. Four dogs with variable experience as therapy dogs participated with their handlers, which were all randomly assigned to the human participants.

To test the hypothesis, the study included self-assessments of the level of anxiety perceived by the patient in previous experiences and the current experience assisted by a dog. Physiological measurements, specifically pulse and blood pressure, were taken at three instances: before, midpoint, and at the end of the intervention.

According to the results obtained, there was a history of dental anxiety in the patients as expected. Aspects such as thinking about going and waiting for the instruments to be prepared for the intervention were the reasons that provoked the most unease. After the intervention, significant differences were found in how the experience of going to and participating in the dental appointment was evaluated: being accompanied by a dog led to a decrease in discomfort. In addition, there was an improvement in how the experience of going to the dentist was perceived.

These results support the working hypothesis proposed in this research: the presence of an animal (dog) decreases the participants’ anxiety, which is already documented in nondental spaces [14]. This fact was corroborated by the decrease in blood pressure (high and low) in the middle of the intervention. For pulse, other physiological measure were used; although it showed a similar behavior—decrease during the intervention—its change did not reach statistical significance.

In another study of 30 patients with intellectual disabilities, ranging in age from 24 and 66 years, who received dental care under sedation, the therapy dog accompanied the patients until the moment at which the anesthesia was induced. Anxiety was measured with the Anxiety Depression and Mood Scale (ADAMS), evaluating the behavior before and after interacting with the therapy dog, and a significant difference (*p* < 0.019) was found in the reduction of anxiety, from moderate levels (M = 4.30, SD = 0.79) to minor levels (M = 2.93, SD = 0.86) [24]. These results coincide with those of the present study, where the physiological data show a decrease in symptoms associated with an anxious state.

While little scientific information has been reported so far on the psychological benefits provided by therapy dogs to patients with dental anxiety, the results are encouraging, and for this reason, it is of great importance to expand these data based on scientific evidence that justifies the use of these non-invasive procedures in people with disabilities, especially those disabilities in which people cannot communicate or express their emotions. In addition to contributing to strengthening and regulating the biosafety protocols of these procedures, patients with disabilities may be systemically more vulnerable, which further research can help address.

This project was based on the theory of learning as an antecedent to how the person will experience a new situation: it situates the dental anxiety experienced by the participants and its acquisition in various ways [1,8,9,10,11], in the face of empirical knowledge that the presence of an animal has effects on perceived anxiety in different spaces and for different reasons [14]. Accordingly, it is observed that there is indeed support for managing dental anxiety using the presence of animals.

Although this approach assumes that a basic requirement for working with dogs is to feel comfortable with them, which can be seen as a determining factor in the outcome, in turn, it considers the cognitive and physiological responses associated with discomfort that can diminish or even lead to abandoning dental treatments with gradual adverse consequences for general health [3,4]. For this reason, apart from expanding the knowledge about the effect of animals on human health, this research opens up the opportunity to consider it as an option to augment existing techniques to support patients with this type of negative reaction [12,13], considering the merging of several disciplines with the same objective.

Regarding the limitations of this pilot study, the small sample size is the main limitation, because a small number of participants does not enable generalizing the results. However, the results demonstrate an improvement in controlling negative emotions such as dental anxiety in people without disabilities, thus promoting the development of broader multidisciplinary studies on the subject by psychologists, orthodontists, and veterinarians. Since it is a simple, non-invasive and economical procedure, AAT is an excellent option for the management of dental anxiety, especially in people who live with dogs on a daily basis and keep them indoors as part of their family. A further limitation has been the acceptance of AAT in the dental environment by health personnel, because when seeking authorization from local health authorities to implement a large-scale project in dental offices, authorization was rejected, citing possible hygiene problems or risk to patients. Thus, research on the subject must be expanded, including an evaluation of possible infections or other health risks for humans and dogs.

Recommendations for the continuity of this project include testing this type of intervention with people with disabilities and with patients who have resorted to sedation to explore its possibility as a less invasive alternative.

## Figures and Tables

**Table 1 animals-09-00512-t001:** Means and standard deviations of the Corah Dental Anxiety Scale (CDAS) [26] scale.

Questions from the CDAS Scale	Mean	SD
If you had to go to the dentist tomorrow, how would you feel?	3.67	1.30
While you are waiting at the dentists for you turn in the dental chair, do you feel?	3.00	1.20
While you are in the dentist’s chair while he gets his drill ready to begin work on your teeth, how do you feel?	3.17	1.26
You are in the dentist’s chair to have your teeth cleaned. While you are waiting and the dentist is getting out the instruments he will use to scrape your teeth around your gums how do you feel?	3.42	1.50

Note: Maximum value per item = 5.

**Table 2 animals-09-00512-t002:** Participant’s perceptions before and after the assisted dental intervention with dogs.

	How Did You Feel About Going to the Dentist?		What Did You Think of Going in with (Name of the Dog)?	Going to the Dentist is…	
	Pre	Post	Post	Pre	Post
Mean	3.2	4.9	5.8	2.2	4.5
Standard deviation	1.1	0.9	0.4	1.4	0.9
Participant 1	4	6	6	1	4
Participant 2	3	5	6	2	5
Participant 3	3	4	6	1	5
Participant 4	5	5	6	4	5
Participant 5	3	6	6	4	5
Participant 6	4	5	6	5	4
Participant 7	1	3	6	1	4
Participant 8	4	5	6	2	3
Participant 8	2	5	5	1	4
Participant 10	4	5	6	2	6
Participant 11	3	6	6	2	5
Participant 12	2	4	5	2	4

Note: Answer range: 1 = It is the worst, 6 = It is the best.

**Table 3 animals-09-00512-t003:** Physiological measurements and the Friedman test before, midpoint, and at the end of the intervention.

Measurement	Before		Midpoint		End		Friedman Test	
	Me	IQR	Me	IQR	Me	IQR	X^2^	*p*
Pulse	84.50	2.54	75.50	1.63	78.50	1.83	5.91	0.052
High Pressure	140.50	2.29	130.50	1.42	142.50	2.29	6.25	0.044
Low Pressure	93.50	2.33	80.00	1.33	94.50	2.33	8.34	0.015

Note: Me = Median, RI = Interquartile range.

**Table 4 animals-09-00512-t004:** Physiological measurements and Wilcoxon ranks before and midpoint.

Measurement	Wilcoxon Ranks	
	Z	*p*
Pulse	2.67	0.007
High Pressure	2.71	0.007
Low Pressure	2.12	0.034

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
