# Peer review of "Dog-Assisted Therapy and Dental Anxiety: A Pilot Study"

_animals, 2019, doi:10.3390/ani9080512_

Round 1
Reviewer 1 Report
This review refers to the submitted manuscript animals-532563 “DOG-ASSISTED THERAPY AND DENTAL ANXIETY”. The topic is timely as there is little information available on how animals may support humans at the dentist, compared to other therapeutic environments (e.g. oncology, post-surgery, rehabilitation). The choice of methodology is appropriate, however, some moderate-major revision is certainly needed prior to publication. Given the small sample, the imbalance in sex distribution and thus, the preliminary nature of the data, adding „a pilot study“ to the title is highly recommended. I suggest moving Table 2 to the Appendix as the frequencies and percentages of the CDAS scale do not seem like a central result of the study. Instead, I suggest adding a new table including the results of the self-designed questions as to how participants felt before and after the dental intervention with a therapy dog. 11 females in their fertile age span participated but differences in basal blood pressure across the female ovarian cycle (e.g. Dunne et al. 1991) were obviously not controlled for, however their impact should at least be mentioned in the discussion. The level of English throughout the manuscript can be improved!
L 37 delete This, add Avoidance behavior
L39 depending on individual coping resources [5,6]
L66 regarding the role of animals limit the dissemination
L69/70 levels of dental anxiety
L88 For this study, a minimum score of 13
L97/98 The questionnaire was designed as a Likert scale with 6 possible answers, the numerical values of which were weighted from 1 (lowest) to 6 (highest). à What exactely did the participants rate on the 6 point scale? Good-Bad? Anxiety? Agreement?
L118 social networks - which?
L128 On the appointed date, the therapy dog
L130 Were participants really encouraged to hold the therapy dog? How is this in line with the current animal welfare debate?
L207/208 that the presence of the dog would decrease
L223 the participants’ anxiety, which
Table 1 caption: Means and standard deviations of the CDAS [26] scale.
Table 2 caption: Frequencies and percentages of the answers given per item of the CDAS [26] scale.
Author Response
Reviewer 1
Dear Reviewer,
We appreciate all your interesting questions and comments. Here you have the answers for each of your questions.
This review refers to the submitted manuscript animals-532563 “DOG-ASSISTED THERAPY AND DENTAL ANXIETY”. The topic is timely as there is little information available on how animals may support humans at the dentist, compared to other therapeutic environments (e.g. oncology, post-surgery, rehabilitation). The choice of methodology is appropriate, however, some moderate-major revision is certainly needed prior to publication. Given the small sample, the imbalance in sex distribution and thus, the preliminary nature of the data, adding „a pilot study“ to the title is highly recommended.
We added “a pilot study” to the title.
I suggest moving Table 2 to the Appendix as the frequencies and percentages of the CDAS scale do not seem like a central result of the study.
We agree and did it
Instead, I suggest adding a new table including the results of the self-designed questions as to how participants felt before and after the dental intervention with a therapy dog.
The table was added, this is now Table 2
11 females in their fertile age span participated but differences in basal blood pressure across the female ovarian cycle (e.g. Dunne et al. 1991) were obviously not controlled for, however their impact should at least be mentioned in the discussion.
The variation of blood pressure during the menstrual cycle is an interesting topic. The reference suggested by the reviewer was reviewed, as well as a meta-analysis and other papers. Dunne et al. (1991) mentioned that previous to their work, changes in blood pressure during the normal menstrual cycle had not been well documented, and those previous studies had given conflicting results. In their study they found that morning blood pressure alters during the normal menstrual cycle, being higher at the onset of menstruation and lower on days 17-26, when compared with the remainder of the cycle.
In the meta-analytic review performed by Riley et al. (1999), some studies are mentioned about it. They affirm that blood pressure changes across menstrual phases.
The documented variations are during the menstrual cycle, however, in the present investigation only one day was measured, no changes in blood pressure were evaluated from one day to the next. The limitation could be when comparing blood pressure between women or among days, without considering their stage of the menstrual cycle. In the present study there was no comparison group, and neither were men compared to women; therefore, we consider that it is not necessary to include this information in the discussion.
Dunne, F. P., Barry, D. G., Ferriss, J. B., Grealy, G., & Murphy, D. (1991). Changes in blood pressure during the normal menstrual cycle. Clinical Science, 81(s25), 515-518.
Riley III, J. L., Robinson, M. E., Wise, E. A., & Price, D. (1999). A meta-analytic review of pain perception across the menstrual cycle. Pain, 81(3), 225-235.
Nevertheless, if reviewers and editor consider this information relevant, we will include it in the discussion.
The level of English throughout the manuscript can be improved!
We appreciate all your corrections
L 37 delete This, add Avoidance behavior
Done
L39 depending on individual coping resources [5,6]
Done
L66 regarding the role of animals limit the dissemination
Done
L69/70 levels of dental anxiety
Done
L88 For this study, a minimum score of 13
Done
L97/98 The questionnaire was designed as a Likert scale with 6 possible answers, the numerical values of which were weighted from 1 (lowest) to 6 (highest). à What exactely did the participants rate on the 6 point scale? Good-Bad? Anxiety? Agreement?
In the questionnaire given to the participant value 6 meant the best experience. For a clearer reading, this information was changed as follows:
The questionnaire was designed as a Likert scale with 6 possible answers, the numerical values of which were weighted from 1 (the worst) to 6 (the best).
L118 social networks - which?
Social networks was changed by Facebook (Now L121)
L128 On the appointed date, the therapy dog
Done (L132)
L130 Were participants really encouraged to hold the therapy dog? How is this in line with the current animal welfare debate?
Patients were instructed to not touch the dog until the dog approached him/her, this information was given at the waiting room, before they enter to the consulting room. At this point, the handler gave this information and some treats to the participant as a first step to get in touch with each other. It is also good to mention that the dogs were previously trained to be near or in the lap of the person the dentist attended. The handlers were instructed if the dog did not want to be petted, they should leave the session, although it never happened.
L133/134 was changed as follows:
(where and how to hold it, so that the dog feels comfortable)
L207/208 that the presence of the dog would decrease
Done (L212)
L223 the participants’ anxiety, which
Done (L228)
Table 1 caption: Means and standard deviations of the CDAS [26] scale.
Done
Table 2 caption: Frequencies and percentages of the answers given per item of the CDAS [26] scale.
Done, this is now Appendix A caption
We appreciate your contributions
The authors

Reviewer 2 Report
Overall, the project is well organized and well-described. The one caveat is that I expected a more formal treatise based on the title. I think including "A Pilot Study" in the title would help readers understand the scope of the manuscript.
One of the common challenges associated with the use of dogs in clinical settings is coordinating availability of the dog team and the patient's calendar. Some detail as to how this was achieved in the study would be helpful.
Were the dog teams volunteers? Were the owners trained in privacy standards, etc.? Were they compensated for participating?
Author Response
Dear Reviewer,
We appreciate all your interesting questions and comments. Here you have the answers for each of your questions.
Overall, the project is well organized and well-described. The one caveat is that I expected a more formal treatise based on the title. I think including "A Pilot Study" in the title would help readers understand the scope of the manuscript.
We added “a pilot study” to the title.
One of the common challenges associated with the use of dogs in clinical settings is coordinating availability of the dog team and the patient's calendar. Some detail as to how this was achieved in the study would be helpful.
All sessions were on Saturday, so patients and volunteer handlers had more time available, volunteer schedules were previously asked and then the appointment was schedule in agreement with the patient.
We added this information “…dates and time available, depending on the availability of the volunteer handlers” (L126), complete paragraph is as follows:
The interested parties received more detailed information about the project, which included the previously described inclusion criteria, dates and time available, depending on the availability of the volunteer handlers, the need to sign an informed consent where the project was made known in greater detail, the indicated procedure was authorized, the videotaping of the session was permitted, and the participant agreed to complete the previously described questionnaires.
Were the dog teams volunteers? Yes
Were the owners trained in privacy standards, etc.? Yes
Were they compensated for participating? No
We added this information (L114):
Canine teams worked as volunteers, handlers were not compensated for participating and were previously trained in the privacy standards and all the necessary details about the dental treatment.
We appreciate your contributions
The authors
